# Coinfection with *Leishmania infantum* and *Toxoplasma gondii* in Domestic Cats from a Region with a High Prevalence of Feline Immunodeficiency Virus

**DOI:** 10.3390/microorganisms12010071

**Published:** 2023-12-29

**Authors:** José Artur Brilhante Bezerra, Amanda Haisi, Gabrielle dos Santos Rocha, Suellen Gonçalves Lima, Arthur Willian de Lima Brasil, Klívio Loreno Raulino Tomaz, Felipe Fornazari, Helio Langoni, João Pessoa Araújo Junior, João Marcelo Azevedo de Paula Antunes, Sérgio Santos de Azevedo

**Affiliations:** 1Centro de Saúde e Tecnologia Rural, Universidade Federal de Campina Grande (UFCG), Av. Universitária, s/n, Santa Cecília, Patos 58708-110, Brazil; artur_brilhante@hotmail.com; 2Instituto de Biotecnologia, Universidade Estadual Paulista (UNESP), Alameda das Tecomarias, s/n, Chácara Capão Bonito, Botucatu 18607-440, Brazil; amanda.haisi@unesp.br (A.H.); joao.pessoa@unesp.br (J.P.A.J.); 3Departamento de Produção Animal e Medicina Veterinária Preventiva, Faculdade de Medicina Veterinária e Zootecnia, Universidade Estadual Paulista (UNESP), Rua Prof. Doutor Walter Mauricio Correa, s/n, Rubião Júnior, Botucatu 18618-681, Brazil; gabrielle.rocha@unesp.br (G.d.S.R.); suellen.goncalves@unesp.br (S.G.L.); fornazarivet@gmail.com (F.F.); hlangoni@fmvz.unesp.br (H.L.); 4Departamento de Morfologia, Universidade Federal da Paraíba (UFPB), Cidade Universitária, s/n, Campus I, Castelo Branco, João Pessoa 58051-900, Brazil; arthurwillian7@yahoo.com.br; 5Hospital Veterinário Jerônimo Dix-Huit Rosado Maia, Universidade Federal Rural do Semi-Árido (UFERSA), Av. Francisco Mota, 572, Costa e Silva, Mossoro 59625-900, Brazil; klivio@ufersa.edu.br

**Keywords:** FIV, feline leishmaniasis, toxoplasmosis, FeLV, epidemiology, *Felis catus*

## Abstract

The aim of this study was to investigate the coinfection of feline retroviruses (feline immunodeficiency virus—FIV, and the feline leukemia virus—FeLV) with *Leishmania infantum* and *Toxoplasma gondii* and the factors associated with these pathogens in domestic cats from Mossoró, a city endemic for canine and human leishmaniasis situated in the semiarid region of Northeast Brazil. Blood samples from 120 cats were collected, and an epidemiological questionnaire was applied to investigate the risk factors associated with the infections. Retroviruses, *L. infantum*, and *T. gondii* infections were assessed using a point-of-care ELISA and quantitative PCR (qPCR), indirect fluorescent antibody test (IFAT) and qPCR, and IFAT, respectively. The overall seroprevalences observed were 35% (95% CI = 27.0–43.8%) for FIV, 0.8% (95% CI = 0.1–4.5%) for FeLV, 25.8% (95% CI = 18.8–34.3%) for *T. gondii*, and 4.2% (95% CI = 1.7–9.3%) for *L. infantum*. Coinfection with FIV and *L. infantum* was observed in 2.5% (3/120) of the assessed cats, while 12.5% (15/120) were coinfected with FIV and *T. gondii*. No significant association was found among the investigated agents (*p* > 0.05). The factors associated with FIV infection in the multivariable analysis were male sex and age above 78 months. The findings of this study demonstrated a high rate of FIV infection in cats from the Brazilian semiarid region and the exposure of these animals to zoonotic and opportunistic agents. Due to the immunosuppressive potential of FIV, cats infected with this retrovirus should be screened for coinfections with *L. infantum* and *T. gondii*, and preventative measures should be adopted.

## 1. Introduction

The feline immunodeficiency virus (FIV) and the feline leukemia virus (FeLV) are among the most important infectious agents affecting domestic cats worldwide [1]. These viruses belong to the *Retroviridae* family and cause a persistent lifelong infection impacting the immunocompetence and the health of the infected animals [2,3]. Both retroviruses are mainly transmitted via saliva, and infected cats can develop a variety of clinical conditions, such as secondary infections due to immunosuppression, chronic inflammatory diseases, bone marrow suppression, and neoplasia, especially lymphomas [2,4].

Due to the impairment of the immune system caused by feline retroviruses, cats infected with FIV and/or FeLV can be more susceptible to opportunistic pathogens, including agents with zoonotic potential, such as *Leishmania infantum* and *Toxoplasma gondii* [5,6]. Feline leishmaniasis is considered an emerging feline disease [7], and retroviral infections have been associated with *L. infantum* coinfection [5], especially in FIV-positive cats [8,9,10,11,12,13], and it has also been associated with FeLV infection and with coinfection by both retroviruses [14].

Regarding toxoplasmosis, it is known that cats infected with FIV can present a higher seroprevalence for *T. gondii* and develop higher titers of antibodies [15,16]. Moreover, the immune dysfunction observed in felines infected by retroviruses can promote the reactivation of cystic stages in chronically infected animals, resulting in clinical disease [16,17], or the development of severe clinical disease in cats acutely infected by this protozoan [18]. Since these agents are zoonotic, they represent a public health concern, and it is extremely important to recognize their relationship with feline retroviruses to provide the proper healthcare management to the infected cats and minimize the risk of transmission of these diseases to other animals and humans.

The prevalence of FIV and FeLV varies widely around the world and is influenced by the characteristics of the studied populations, such as the geographic area evaluated, lifestyle of the cats, health status, populational density, and level of health care provided by the owners [4]. Previous studies demonstrated that infection by FIV in the Brazilian semiarid region is frequent, with prevalence varying from 24 to 27.5%, while a lower prevalence is observed for FeLV, from 0.9 to 3.29% [19,20].

Few studies have been conducted in Brazil on the coinfection between retroviruses and zoonotic pathogens [19,20,21,22]. Thus, the aim of this study was to investigate the coinfection of FIV and FeLV with *L. infantum* and *T. gondii*, and the factors associated with these pathogens in domestic cats from Mossoró, a city endemic for human and canine leishmaniasis that is situated in the semiarid region of Northeast Brazil.

## 2. Materials and Methods

### 2.1. Study Area and Ethical Statements

The study was conducted in the municipality of Mossoró, state of Rio Grande do Norte, Northeastern Brazil (5°100′ S, 37°100′ W) (Figure 1). The climate of the region is semiarid, and the average temperature is 27.4 °C, with an average relative humidity of 68.9%. Very irregular annual rainfall is observed, with an average of 673.9 mm and a drought period of 6 to 8 months [23].

The Ethics Committee on Animals’ Use (CEUA) of the Universidade Federal Rural do Semi-Árido (UFERSA) approved the experimental protocols and the procedures used for animal care (Process n° 23091.008147/2017-28).

### 2.2. Sampling and Biological Sample Collection

The minimum number of animals to be sampled was determined using simple random sampling:$$n=\frac{Z^{2}\times P(1-P)}{d^{2}}$$

*n* = sampling number.

*Z* = normal distribution value for the 95% confidence level.

*P* = expected prevalence of 50% (sample maximization).

*d* = 10% absolute error.

To perform adjustments for finite populations, the following formula was applied:$$n_{adjus}=\frac{N\times n}{N-n}$$

*n_adjus_* = adjusted sample size.

*N* = total population size.

*n* = initial sample size.

The adjustment of the population sample size took into account the cat-to-human ratio of 19.33:1 that was described by Canatto et al. [24]. The minimum number of animals to participate in the study was 96. However, 120 samples were collected, for safety.

A cross-sectional study was performed from March to September 2021. Overall, 120 cats older than 6 months of age, treated at the Veterinary Hospital Jerônimo Dix-Huit Rosado Maia from UFERSA, were selected for this study. The selection of animals with ages more than 6 months was made to minimize the risk of maternal antibodies interfering with the FIV antibody test. No probabilistic criteria were used for the selection of the animals. To participate in the study, the cat’s owners had to sign a free and informed consent form.

The cats were physically examined, and data related to each animal and its clinical status were recorded in individual medical charts. Blood samples of 5 mL were collected from the jugular or cephalic vein using a sterile 23 G intravenous butterfly needle and a 5 mL syringe. The samples were put into EDTA and gel and clot activator tubes. Samples without anticoagulants were centrifuged for serum obtainment. Whole blood and serum samples were stored at −80 °C prior to serological and molecular assays.

An epidemiological questionnaire was applied to the cat’s owners. The variables were analyzed, and the respective categories were as follows: sex (male, female), age (up to 18 months, between 19 and 78 months, greater than 78 months), neuter status (neutered, not neutered), outdoor access (yes, no), type of food (commercial feed, homemade food), fed with raw meat (yes, no), locale of defecation (litter box, outside the house), history of fighting with stray cats (yes, no), presence of stray cat colonies in the neighborhood (yes, no).

### 2.3. Serology for FIV and FeLV

Immediately after the blood collection, a point-of-care enzyme-linked immunosorbent assay (ELISA) (SNAP FIV/FeLV Combo Test, IDEXX Laboratories, Westbrook, ME, USA) was performed to investigate IgG anti-FIV antibodies and p27 FeLV antigens. The test was executed following the manufacturer’s recommendations, using whole blood samples. The SNAP FIV/FeLV Combo Test presents a sensitivity of 100% and specificity of 100% for FeLV detection and a sensitivity of 97.9% and specificity of 99% for FIV detection [25].

### 2.4. Serology for Toxoplasma gondii and Leishmania spp.

The serological tests for *T. gondii* and *Leishmania* spp. were performed at the Zoonosis Diagnosis Service of the São Paulo State University (UNESP), FMVZ-UNESP, Campus of Botucatu, SP, Brazil. Serum samples were tested for specific IgG and IgM antibodies anti-*T. gondi* and anti-*Leishmania* spp. using the indirect fluorescent antibody test (IFAT) [26]. For toxoplasmosis investigation, serial serum dilutions of 1:16, 1:64, 1:256, 1:1024, and 1:4096 were performed in pH 7.2 phosphate-buffered saline solution (PBS). Immunofluorescence slides were previously sensitized with 0.1% formaldehyde to inactivated tachyzoites of *T. gondii* (RH strain). For *Leishmania* spp., serial serum dilutions of 1:40, 1:80, 1:160, 1:320, and 1:640 were prepared in PBS, and the immunofluorescence slides were sensitized with *Leishmania major* promastigotes obtained from cultures maintained in liver infusion tryptose (LIT) and Neal, Novy, Nicolle (NNN) media. Commercial anti-IgG or anti-IgM antibodies specific to cats, conjugated with fluorescein isothiocyanate (SIGMA-Aldrich, San Luis, MO, USA), were used as secondary antibodies. Slides were examined using a fluorescence microscope at magnification of 40× (Scope.A1; ZEISS, Oberkochen, Germany), and positive and negative controls were used in all slides. The cut-off adopted for *T. gondii* was the dilution of 1:16, while for *Leishmania* spp. the dilution of 1:40 was the cut-off value. After checking the control slides, the highest dilution of the serum, for which complete fluorescence occurred at the border of at least 50% of the antigens, was considered.

### 2.5. Molecular Analyses

Molecular analyses were performed at the Institute of Biotechnology (IBTEC) of UNESP, campus of Botucatu, SP, Brazil. Total nucleic acid was extracted from all whole blood samples using the commercial kit MagMAX™ CORE Nucleic Acid Purification Kit (Thermo Fisher Scientific, Waltham, MA, USA), according to the manufacturer’s instructions. After the extraction, DNA were quantified in a spectrophotometer (NanoDrop™ Lite Spectrophotometer, Thermo Fisher Scientific, Waltham, MA, USA). Extracted samples were stored in DNAse- and RNAse-free microtubes at −80 °C prior to amplification via quantitative polymerase chain reaction (qPCR).

A qPCR was performed to identify the presence of FIV, FeLV, and *L. infantum* infection using SYBR Green detection. For the detection of FIV and FeLV, each qPCR assay included 0.4 µL (10 µM) of each specific primer: forward FIV_gag_F2 5′-ATGGGGAAYGGACAGGGGCGAGA and reverse FIV_gag_R2 3′-TCTGGTATRTCACCAGGTTCTCGTCCTGTA [27]; and forward FFeLV-U3 5′-AACAGCAGAAGTTTCAAGGCC-3′ and reverse RFeLV-U3 5′-TTATAGCAGAAAGCGCGCG-3′ [28], respectively. The qPCRs for FIV and FeLV were performed using 10 µL of 2 × SYBR Green PCR master mix (Promega GoTaq qPCR Master Mix, Madison, WI, USA), 2 × GoTaq^®^ qPCR Master Mix (Promega, Madison, WI, USA), 4.2 µL of nuclease-free water, and 5 µL of total nucleic acid. A cat blood sample positive for FIV and FeLV and nuclease-free water were used as positive and negative controls, respectively.

The qPCR for the detection of *L. infantum* was performed using the previously described primers [29] targeting the kinetoplast DNA minicircle sequence (kDNA). Reactions were performed at a final volume of 20 μL, containing 0.4 μL (10 µM) of each primer: kDNA-F 5′-GGCGTTCTGCAAAATCGGAAAA-3′, Linf kDNA-R 5′-CCGATTTTTGGCATTTTTGGTCGAT-3′, 10 μL of 2× GoTaq^®^ qPCR Master Mix (Promega, Madison, WI, USA), 4.8 μL of nuclease-free water, and 4 µL of total nucleic acid. A positive sample for *L. infantum* from a dog and nuclease-free water were used as positive and negative controls, respectively.

All qPCR runs used the same cycling conditions, including an initial denaturation at 95 °C for 5 min, followed by 40 cycles at 95 °C for 15 s and also at 60 °C for 1 min, followed by the melting curve using the AriaMX Real-Time PCR system (Agilent, Santa Clara, CA, USA). Samples were considered positive for FIV if the cycle threshold (Ct) ≤ 36 and positive for FeLV and *L. infantum* if Ct ≤ 34.

### 2.6. Statistical Analyses

Data were tabulated in an electronic sheet. Independent variables collected from the epidemiological questionnaires and their respective categories were crossed with the test results for the different pathogens (positive, negative) using the Chi-square test or Fisher’s exact test to select the variables with significant differences. Subsequently, variables that presented *p*-value ≤ 0.2 were selected for multivariable analysis using Poisson regression. To verify any possible collinearity between the variables, Pearson’s correlation was applied, and the variables that presented a correlation coefficient > 0.9 were excluded, according to biological plausibility. The significance of the final model was verified via Omnibus test. To investigate the association between the infectious agents, the Chi-square test or Fisher’s exact test was used. The data were analyzed using SPSS 23 for Mac, at a significance level of 5%.

## 3. Results

The study population comprised 120 cats, of which 37 (30.8%) were females and 83 (69.2%) were males, 42 (35.0%) were healthy and 78 (65.0%) were sick, with ages varying from 6 months to 18 years. All animals were crossbreed. The sex, age, and clinical status and the infection and coinfection rates observed in the present study are detailed in Table 1.

Using ELISA, 41 (34.2%; 95% CI = 25.7–42.7%) felines were positive for FIV antibodies, while only one animal (0.8%; 95% CI = 0.1–4.5%) was positive for the FeLV antigen, and it was also coinfected with FIV. According to the qPCR assay, 33 (27.5%; 95% CI = 19.5–35.5%) animals were positive for FIV proviral DNA. None of the sampled cats were found to be positive for FeLV according to the qPCR. Combining the results obtained in ELISA and qPCR, 42 (35.0%; 95% CI = 27.0–43.8%) animals were infected with FIV in this study. The Cohen’s κ-coefficient for ELISA and qPCR in detecting FIV infection was κ = 0.81 (*p* < 0.001), demonstrating a strong agreement between the methods used for FIV diagnosis. The variables sex, age, outdoor access, contact with stray cats, history of fighting with stray cats, and defecation outside the house were associated with FIV positivity in the univariable analysis (*p* ≤ 0.2). The multivariable model of Poisson regression found male sex (prevalence ratio [PR] = 2.9 (95% CI: 1.1–7.8)] and age > 78 months (PR = 2.8; 95% CI: 2.8–6.9) as predictors of a positive result for FIV (*p* < 0.05) (Table 2). The final model presented statistical significance (Omnibus test: Chi-square = 16.532; degrees of freedom = 7; *p* = 0.021).

Regarding the tests that investigated *L. infantum* infection, 4 (3.3%; 95% CI = 1.3–8.2%) cats were positive for IgG antibodies based on IFAT, with titers varying from 1:40 to 1:160. Anti-*Leishmania* IgM was not detected in any of the studied cats. Based on the qPCR assay, kDNA of *L. infantum* was detected in only one animal (0.8%; 95% CI = 0.1–4.5%). The overall rate of *Leishmania* infection was 4.2% (95% CI = 1.7–9.3%). Of these, three were asymptomatic, and two were presenting clinical signs. Sixty percent (3/5) of the cats that tested positive for *Leishmania* using at least one of the laboratory methods were coinfected with FIV, but no statistical association was observed between these two agents (*p* = 0.779). Due to the low number *Leishmania*-positive animals, no tests were applied to assess statistical associations with this pathogen. Descriptive data of the cats positive for *Leishmania* are shown in Table 3.

Using IFAT, IgG antibodies anti-*T. gondii* were observed in 31 (28.5%; 95% CI = 18.8%–34.3%) cats, with titers varying from 1:16 to 1:1024. Five (4.2%) animals presented IgM antibodies anti-*T. gondii*, with titers varying from 1:16 to 1:64 (Table 2). All the IgM-positive cats were also positive for IgG, with the IgM titers lower than those observed for IgG. No significant association was observed in any of the factors evaluated for *T. gondii* infection (*p* > 0.2). Although 48.39% (15/31) of the cats seropositive for *T. gondii* were coinfected with FIV, no statistical association was found between the positivity for these two pathogens (*p* = 0.053). Only one cat presented triple coinfection with FIV, *L. infantum*, and *T. gondii*. Descriptive data of the cats positive for *T. gondii* are shown in Table 4.

## 4. Discussion

The present study described for the first time the coinfection of FIV with *L. infantum* and *T. gondii* in domestic cats from Mossoró city, which is situated in the semiarid region of Northeastern Brazil. Although no statistical association was observed between the occurrence of these pathogens, our findings demonstrated that FIV-positive cats in the studied region are being exposed to zoonotic pathogens. As the FIV causes progressive immune dysfunction in the cats, the coinfection with *L. infantum* and *T. gondii* might favor the development of severe clinical disease [5,17,18], in addition to the risk of these felines acting as a source of infection for other animals and humans.

The prevalence of 35.0% observed for FIV in this study is among the highest described in surveys carried out in Brazil. Conversely, FeLV was diagnosed in only one animal. These findings reinforce previous published data, which described a high prevalence of FIV and a low prevalence of FeLV in cats from the semiarid region [19,20]. This high frequency of FIV reflects the lifestyle of the cats raised in the studied area. For example, our results demonstrated that 55.8% of the sampled cats had outdoor access, 75.8% had contact with stray cats, and 60.0% had history of fighting with stray cats, and all these variables were statistically associated with FIV infection in the univariable analysis. These factors increase the exposure of susceptible animals to potential FIV sources of infection and justify the high rate of positivity for this retrovirus observed in the evaluated region [1].

The multivariable analysis found that male cats had a FIV prevalence 290% higher than the females, and cats with an age greater than 78 months had a prevalence 280% higher than the animals with age up to 24 months. Male cats are frequently involved in fights due to territorialism and disputes for females in estrus, and for these reasons males are a risk group for FIV infection [20,30,31]. Regarding age, previous authors observed the increase in FIV prevalence with advancing age [32,33], corroborating the results described here.

The prevalence of *Leishmania* infection observed in the present study was 4.2%, combining the results of IFAT with qPCR. No agreement was observed between IFAT and qPCR in the diagnosis of leishmaniosis. Feline leishmaniasis diagnosis is challenging because cats are considered more resistant to infection than dogs, generally develop lower titers of antibodies [5], and when different diagnostic methodologies are used, e.g., cytological and molecular tests using different tissue samples in association with serological tests, disagreements among the tests and types of samples are usually observed [8,34,35]. Therefore, the combination of different tests and different tissue samples, whenever possible, is highly recommended to minimize false-negative results.

The presence of anti-*Leishmania* IgM has been poorly investigated in cats [34]. In dogs, specific IgM antibodies were detected in natural and experimental infections less frequently than IgG antibodies [36,37,38,39], and they are not considered a marker of acute infection [38]. Our results did not find anti-*Leishmania* IgM antibodies in the sampled cats. Further studies are necessary to clarify the value of IgM investigation on feline leishmaniasis diagnosis.

The observed rate of positivity for *Leishmania* using IFAT was 3.3% and 0.8% using blood qPCR. A previous survey carried out in the same city found a seroprevalence of 15.38% for this protozoan using the same serological technique, but none of the tested animals was positive based on the PCR assay using blood samples [19]. It is important to highlight that this is the first study to report *L. infantum* kDNA detection in cats in the assessed area. The city of Mossoró and the state of Rio Grande do Norte are endemic for canine and human leishmaniasis [40,41], and the identification of infected cats alerts us to the possible participation of these animals in the epidemiological chain of the disease.

Among the five cats infected with *Leishmania*, coinfection with FIV was observed in three animals, but no statistical association was observed between these infectious agents. The statistical inferences regarding *Leishmania* infection in this study were impaired by the low number of positive animals, but the association between this protozoan and FIV has been reported by different authors [8,9,10,11,13], similarly to the association described for leishmaniasis and HIV infection in humans [42]. Since FIV causes immune dysfunction, measures should be adopted in FIV-positive animals to prevent opportunistic and zoonotic diseases, such as *Leishmania* infection, especially in endemic areas for leishmaniasis.

Our results demonstrated a seroprevalence of 28.5% for *T. gondii* using IFAT. The members of the *Felidae* family are the only definitive hosts of this protozoan, and domestic cats are considered key hosts for the maintenance and transmission of this parasite in urban environments [43]. No association was observed with the risk factors evaluated for *T. gondii* seropositivity in this study, but *Toxoplasma* sp. infection has been associated with the different features of the studied cat populations, such as the cat’s lifestyle, sex, age, breed, and neuter status, which might influence predation and social behavior [44,45,46]. The seroprevalence observed in this study was lower than the mean seroprevalence described in recent systematic reviews with feline toxoplasmosis in Brazil and across the world, which were 35.9% and 35%, respectively [47,48]. Feitosa et al. [20] investigating anti-*T. gondii* antibodies in domestic cats from Sousa city, also located in the semiarid region, found a higher prevalence, with 53.4% positivity.

*T. gondii* infection is common in domestic cats, but the clinical disease is rare. Toxoplasmosis in cats is associated with immunosuppression and usually affects the central nervous system (CNS), muscles, lungs, and eyes [49]. In this study, most of the seropositive cats for *T. gondii* were sick, but none of them was manifesting clinical signs related to toxoplasmosis.

Although no significant association was observed between positivity for *T. gondii* and FIV, almost 50% (15/31) of the animals with *T. gondii* antibodies were coinfected with FIV, comprising 12.5% of the assessed population. This high coinfection rate is worrying because FIV-positive cats are immunosuppressed and can develop severe clinical toxoplasmosis after primary infection or due to reactivation of tissue cysts containing bradyzoites in chronically infected cats [16,18]. The animals of this research did not present serological evidence of reactivation of *T. gondii* infection, but since FIV causes a progressive immune dysfunction, advancing immunosuppression might determine the development of clinical disease in coinfected cats [50]. Comparatively, in humans, toxoplasmosis is considered the most common opportunistic cerebral infection and a frequent cause of mortality in HIV-positive patients [51,52].

## 5. Conclusions

The findings of this study demonstrated a high prevalence of FIV infection in cats from the Brazilian semiarid region, with association between male sex and age greater than 78 months in the multivariable analysis. Moreover, regarding the One Health perspective, the presence of *L. infantum* and *T. gondii* in FIV-positive animals demonstrates a major concern for public health, since these agents are opportunistic, and infected felines could act as possible reservoirs for these zoonoses. In conclusion, since the cats from the studied region present an alarming rate of FIV infection and due to the immunosuppressive potential of this retrovirus, infected animals should be screened for coinfections with *T. gondii* and *L. infantum*, and preventative measures should be adopted.

## Figures and Tables

**Figure 1 microorganisms-12-00071-f001:**
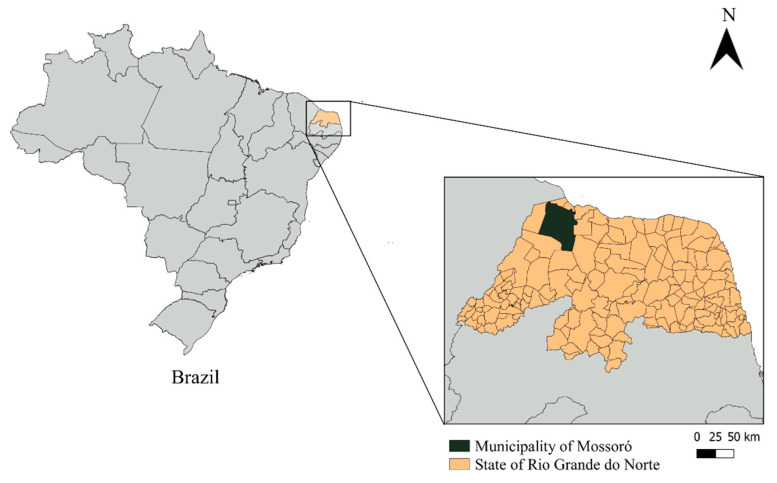
Geographical location of the municipality of Mossoró, state of Rio Grande do Norte, Brazil, where the study was conducted.

**Table 1 microorganisms-12-00071-t001:** Total and percentage of cats infected by feline immunodeficiency virus (FIV), feline leukemia virus (FeLV), *Leishmania infantum*, and *Toxoplasma gondii*, from Brazil, according to sex, age and clinical status.

		Infected Cats							
	Number of Tested Cats	FIV ^1^ (%)		FeLV (%)		*L. infantum*^2^ (%)		*T. gondii* (%)	
Total population	120	42 (35.0)		1 (0.8)		5 (4.2)		31 (25.8)	
Sex									
Female	37	5 (13.5)		0 (0)		3 (8.1)		8 (21.6)	
Male	83	37 (44.6)		1 (1.2)		2 (2.4)		23 (27.7)	
Age (months)									
≤24	32	5 (15.6)		0 (0)		0 (0)		6 (18.75)	
25–78	58	21 (36.2)		0 (0)		2 (3.4)		12 (20.7)	
>78	30	15 (50.0)		1 (3.3)		3 (10)		13 (43.3)	
Clinical status									
Healthy	42	12 (28.6)		0 (0)		2 (4.8)		6 (14.8)	
Sick	78	30 (38.5)		1 (0.8)		3 (3.8)		25 (32.05)	
		Coinfection with FIV							
		FeLV (%)	*L. infantum* (%)		*T. gondii* (%)		*T. gondii* + FeLV (%)		*T. gondii* + *L. infantum* (%)
Total population	120	1 (0.83)	3 (2.5)		15 (12.5)		1 (0.83)		1 (0.83)
Sex									
Female	37	0 (0)	2 (5.4)		1 (2.7)		0 (0)		0 (0)
Male	83	1 (1.2)	1 (1.2)		14 (15.7)		1 (1.2)		1 (1.2)
Age (months)									
≤24	32	0 (0)	0 (0)		1 (3.12)		0 (0)		0 (0)
25–78	58	0 (0)	2 (3.4)		5 (8.6)		0 (0)		0 (0)
>78	30	1 (3.3)	1 (3.3)		9 (30.0)		1 (3.3)		1 (3.3)
Clinical status									
Healthy	42	0 (0)	1 (2.4)		2 (4.8)		0 (0)		0 (0)
Sick	78	1 (1.3)	2 (2.6)		13 (16.7)		1 (1.3)		1 (1.3)

^1^ ELISA + qPCR. ^2^ IFAT + qPCR.

**Table 2 microorganisms-12-00071-t002:** Univariable and multivariable analyses of factors associated with positivity for feline immunodeficiency virus based on ELISA and/or qPCR in domestic cats, Brazil (*n* = 120).

Factor	Group	Total	Positive Cats (%)	Univariable Analysis	Multivariable Analysis	
				*p*	*p*	PR [95% CI]
Sex	Female	37	5 (13.5)	0.001	-	-
	Male	83	37 (44.6)		0.031	2.9 [1.1–7.8]
Age (months)	≤24	32	5 (15.6)	0.015	-	-
	25–78	58	22 (37.9)			
	>78	30	15 (50.0)		0.023	2.8 [2.8–6.9]
Outdoor access	No	53	12 (22.6)	0.012	-	-
	Yes	67	30 (44.8)		-	-
Contact with stray cats	No	29	7 (24.1)	0.159	-	-
	Yes	91	35 (38.5)		-	-
History of fighting with stray cats	No	48	9 (18.8)	0.002	-	-
	Yes	72	33 (45.2)		-	-
Defecation outside the house	No	47	9 (19.1)	0.003	-	-
	Yes	73	33 (45.2)		-	-

**Table 3 microorganisms-12-00071-t003:** Descriptive characteristics of cats positive for *Leishmania infantum*.

Animal	Sex	Age (Months)	*Leishmania infantum* Status		FIV-Status (ELISA/qPCR)	IFAT for *Toxoplasma gondii* (Titer)	Clinical Signs
			IFAT (Titer)	qPCR			
#17	Female	96	+(1:80)	-	(−/−)	-	Asymptomatic
#63	Male	120	+(1:160)	-	(+/+)	+(1:256)	Periodontal disease, chronic upper respiratory infection
#69	Male	216	+(1:80)	-	(−/−)	+(1:256)	Asymptomatic
#71	Female	48	-	+	(−/+)	-	Bilateral blepharitis
#101	Female	48	+(1:40)	-	(+/+)	-	Asymptomatic

**Table 4 microorganisms-12-00071-t004:** Descriptive characteristics of cats positive for *Toxoplasma gondii*.

Animal	Sex	Age (Months)	IFAT for *Toxoplasma gondii*		FIV Status (ELISA/qPCR)	Clinical Signs
			IgG (Titer)	IgM (Titer)		
#1	Male	120	+(1:64)	−	(+/+)	Uveitis, chronic kidney disease
#9	Female	120	+(1:16)	−	(−/−)	Mammary neoplasia
#10	Female	156	+(1:16)	−	(−/−)	Bacterial conjuntivitis
#19	Female	24	+(1:256)	−	(−/−)	Rectal prolapse
#20	Male	108	+(1:16)	−	(+/+)	Asymptomatic
#22	Male	36	+(1:64)	−	(+/−)	Asymptomatic
#26	Male	60	+(1:16)	−	(−/−)	Upper respiratory infection
#28	Female	36	+(1:16)	−	(−/−)	Asymptomatic
#40	Male	132	+(1:256)	−	(−/−)	Cutaneous squamous cell carcnimoma
#41	Male	120	+(1:64)	−	(+/+)	Chronic kidney disease
#48	Male	120	+(1:1024)	−	(+/−)	Asymptomatic
#50	Male	96	+(1:256)	−	(−/−)	Asymptomatic
#58	Female	24	+(1:1024)	−	(+/−)	Lymph node enlargement, gingivitis, diarrhea
#59	Male	36	+(1:64)	−	(−/−)	Upper respiratory infection
#63	Male	120	+(1:256)	−	(+/+)	Periodontal disease, chronic upper respiratory infection
#69	Male	216	+(1:256)	−	(−/−)	Asymptomatic
#73	Male	36	+(1:64)	−	(+/+)	Chronic gingivostomatitis, upper respiratory infection
#80	Male	17	+(1:16)	−	(−/−)	Chronic gingivostomatitis
#81	Male	120	+(1:16)	−	(+/+)	Chronic gingivostomatitis
#83	Female	48	+(1:64)	−	(−/−)	Asymptomatic
#84	Female	48	+(1:256)	−	(−/−)	Asymptomatic
#89	Male	60	+(1:16)	−	(−/−)	Upper respiratory infection
#92	Male	16	+(1:64)	−	(−/−)	Upper respiratory infection
#94	Male	29	+(1:16)	−	(+/+)	Chronic respiratory infection
#97	Female	6	+(1:256)	+(1:64)	(−/−)	Diarrhea
#99	Male	84	+(1:256)	+(1:16)	(+/+)	Multicentric lymphoma
#102	Male	24	+(1:64)	−	(−/−)	Chronic gingivostomatitis
#103	Male	48	+(1:256)	−	(+/+)	Chronic gingivostomatitis
#104	Male	120	+(1:64)	+(1:16)	(+/+)	Cutaneous squamous cell carcinoma
#109	Male	72	+(1:64))	+(1:16)	(+/+)	Chronic gingivostomatitis
#114	Male	84	+(1:256)	+(1:64)	(+/+)	Chronic gingivostomatitis

## Data Availability

Data are contained within the article.

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
