# Peer review of "Coinfection with Leishmania infantum and Toxoplasma gondii in Domestic Cats from a Region with a High Prevalence of Feline Immunodeficiency Virus"

_microorganisms, 2023, doi:10.3390/microorganisms12010071_

Round 1
Reviewer 1 Report
Comments and Suggestions for Authors
The conclusions could be improved in a ONE HEALTH perspective.
It establishs the correlation between FIV and feline leishmaniasis and Toxoplasma gondii.
The topic is not original but relevant and, above all, it addresses the increasingly topical issue of the circulation of feline leishmaniasis.
The topic of the spread of feline Leishmaniasis is current and the work offers new knowledge in the relevant territory.
The methodology used appears adequate both from the point of view of indirect diagnostics (IFAT) and from the molecular one (qPCR).
The conclusions, although obvious, are adequate to the arguments presented.
References are adequate to support the argument and represents the main bibliographic sources present in the literature on feline pathology.
Not necessary further comments or findings on the tables and figures that are consistent with the work carried out.
Thus, I recommend to accept it for publication.
Reviewer 2 Report
Comments and Suggestions for Authors
The selected topic refers to an important part of feline infectious pathology, FIV and FeLV, and zoonotic pathogens originating from cats.
The methods are standardized and suitable, and both the methods and results are well-detailed. The results were analysed by suitable statistical methods
However, the overall presentation for the abstract, introduction and some paragraphs of the discussion has a basic scientific level and the authors are recommended to perform several modifications:
The abstract - the authors should mention the use of molecular methods and also emphasize the novelty
The novelty of the study is not (clearly) presented in the introduction, lines 80-84 mention twice "...the semiarid region of..." and "few studies" that were previously reported and this is not sufficient to present novelty
Are there any epidemiological particularities for this region for any involved species (cats, human)?
Is there any relevant data obtained by molecular methods to underline the study's novelty ?
The authors can not mention "this is the first study", but compared to the previous ones in the region /country " are there any particularities?
https://www.sciencedirect.com/science/article/abs/pii/S0304401712000131?via%3Dihub
2. The introduction is well-structured, but the scientific level should be upgraded, the current form resembles textbooks.
lines 61 -64 should be rephrased to avoid words repetitions.
3. Discussion is mostly focused on the seroprevalence and risk factors, several paragraphs develop ideas that are not clearly supported by the results - "no statistical association", "might" etc
The pathogenesis related to FIV and FeLV infection is known, the induced immunodeficiency increases cats susceptibility to any infections and numerous studies reported the coinfections
As for the abstract and introduction, the authors do not underline the novelty of the study
Comments on the Quality of English Language
Moderate editing of the English language is required, mostly syntax corrections
Round 2
Reviewer 2 Report
Comments and Suggestions for Authors
The authors performed the required modifications